# Childhood Trauma Predicts Less Remission from PTSD among Patients with Co-Occurring Alcohol Use Disorder and PTSD

**DOI:** 10.3390/jcm9072054

**Published:** 2020-06-30

**Authors:** Paul Brunault, Kevin Lebigre, Fatima Idbrik, Damien Maugé, Philippe Adam, Servane Barrault, Grégoire Baudin, Robert Courtois, Hussein El Ayoubi, Marie Grall-Bronnec, Coraline Hingray, Nicolas Ballon, Wissam El-Hage

**Affiliations:** 1CHRU de Tours, Service d’Addictologie Universitaire, Équipe de Liaison et de Soins en Addictologie, 37044 Tours, France; k.lebigre@chu-tours.fr (K.L.); d.mauge@chu-tours.fr (D.M.); hussein.elayoubi@univ-tours.fr (H.E.A.); nicolas.ballon@univ-tours.fr (N.B.); 2CHRU de Tours, Clinique Psychiatrique Universitaire, 37044 Tours, France; robert.courtois@univ-tours.fr (R.C.); wissam.elhage@univ-tours.fr (W.E.-H.); 3UMR 1253, iBrain, Université de Tours, Inserm, 37020 Tours, France; 4Qualipsy EE 1901, Université de Tours, 37020 Tours, France; servane.barrault@univ-tours.fr (S.B.); gregoire.baudin@u-paris.fr (G.B.); 5Soins de Suite et de Réadaptation en Addictologie “Le Courbat”, 37460 Le Liège, France; medecin@lecourbat.fr (F.I.); padam@sante-escale41.fr (P.A.); 6CHRU de Tours, Centre de Soins d’Accompagnement et de Prévention en Addictologie CSAPA-37, 37044 Tours, France; 7Laboratory of Psychopathology and Health Processes EA 4057, University Paris Descartes, Sorbonne Paris Cité, 92100 Boulogne-Billancourt, France; 8Addictology and Psychiatry Department, Hôpital Saint Jacques, University Hospital of Nantes, 85 rue Saint Jacques, Cedex 1, 44093 Nantes, France; marie.bronnec@chu-nantes.fr; 9Inserm, SPHERE U1246 methodS in Patients-Centered Outcomes and HEalth ResEarch, Université de Nantes, Université de Tours, 22 boulevard Benoni Goullin, 44200 Nantes, France; 10Pôle Universitaire du Grand Nancy, Centre Psychothérapique de Nancy, 54520 Laxou, France; c.hingray@chu-nancy.fr

**Keywords:** Substance-use disorder, substance-related disorders, alcohol use disorder, post-traumatic stress disorder, dual disorders, childhood trauma, psychiatric disorders, rehabilitation centers, impulsive behavior, addictive disorders

## Abstract

Post-traumatic stress disorder (PTSD) is highly prevalent among patients hospitalized for an alcohol use disorder (AUD). Hospitalization can improve PTSD and AUD outcomes in some but not all patients, but we lack data on the baseline predictors of PTSD non-remission. This study aimed to determine the baseline risk factors for non-remitted PTSD in patients hospitalized for an AUD. Of 298 AUD inpatients recruited in a rehabilitation center (Le Courbat, France), we included 91 AUD inpatients with a co-occurring PTSD and a longitudinal assessment at baseline (T1) and before discharge (T2: 8 weeks later). Patients were assessed for PTSD diagnosis/severity (PCL-5=PTSD Checklist for DSM-5), different types of trauma including childhood trauma (LEC-5=Life Events Checklist for DSM-5/CTQ-SF=Childhood Trauma Questionnaire, Short-Form), and AUD diagnosis/severity (clinical interview/AUDIT=Alcohol Use Disorders Identification Test). Rate of PTSD remission between T1 and T2 was 74.1%. Non-remitted PTSD at T2 was associated with a history of childhood trauma (physical, emotional or sexual abuse, physical negligence), but not with other types of trauma experienced, nor baseline PTSD or AUD severity. Among patients hospitalized for an AUD with co-occurring PTSD, PTSD remission was more strongly related to the existence of childhood trauma than to AUD or PTSD severity at admission. These patients should be systematically screened for childhood trauma in order to tailor evidence-based interventions.

## 1. Introduction

Alcohol use disorders (AUD), which are characterized by compulsive alcohol use and loss of control over alcohol intake [1], are a major public health problem worldwide [2]. AUD are among the most prevalent psychiatric disorders globally, affecting 8.6% (95% Confidence Interval=CI: 8.1–9.1) of men and 1.7% (95% CI: 1.6–1.9) of women (total point estimate: 5.1%, 95% CI: 4.9–5.4) [1], and they represent a significant health, social, and economic burden to Western societies [3]. AUD are often associated with other addictive and psychiatric disorders, and the co-occurrence of other addictive and psychiatric disorders (i.e., dual diagnosis) is a challenge for clinicians given their prevalence and poor outcome [4]. One of the most prevalent psychiatric disorders associated with AUD is post-traumatic stress disorder (PTSD); while the prevalence of PTSD ranges from 4.8% to 8% in the overall population [5,6], it is much higher in people with AUD and is estimated to be between 20% and 39% [7,8]. For these patients, a PTSD diagnosis is associated with a poorer AUD outcome, as well as a higher rate of hospitalization and a more severe social impairment [5,9,10].

Given the high prevalence of PTSD in patients with AUD and the major burden it represents, a better understanding of the variables associated with greater AUD or PTSD severity is of paramount importance to determine the most effective interventions for patients with this dual diagnosis. As the factors found to be associated with AUD or PTSD severity in cross-sectional studies may not necessarily be linked to poorer long-term outcomes, longitudinal studies are needed. One interesting research area is to identify the specific predictors of the course of PTSD and particularly of poorer remission rates. Based on the hypothesis that PTSD could be a causal risk or maintenance factor for AUD (i.e., improvement in PTSD associated with lower alcohol dependence) [11], identification of the factors associated with poor PTSD outcomes using a longitudinal approach could improve our ability to identify the patients who would benefit from tailor-made interventions, and ultimately improve AUD outcome [11].

In general, different predictors of poor PTSD outcomes have been identified, including being female, lower socio-economic status, childhood trauma, lifetime and childhood sexual trauma, PTSD severity or type (i.e., more severe PTSD symptoms at baseline, high combat exposure, trauma severity), greater number of stressors prior to trauma, other comorbid psychiatric disorders (e.g., mood and anxiety disorders, personality disorders), and social factors (e.g., lack of social support) [12,13,14,15]. When focusing on the specific population of patients with an AUD, studies have demonstrated that PTSD was associated with avoidance symptoms [16]. Childhood trauma, which is very prevalent in patients with an AUD [17,18], is strongly associated with the development of AUD and PTSD, and could play a central role in maintaining the association between the two disorders [19,20]. One possible explanation of the association between childhood trauma, PTSD, and AUD refers to Bowlby’s theoretical framework and attachment theory [21]. According to Bowlby, attachment can be understood within an evolutionary context in that the caregiver (i.e., attachment figure) provides safety and security to the infant. Bowlby postulated that children who perceive their attachment figure as nearby, accessible, and attentive may be more likely to experience a secure attachment bond, while children who do not perceive their attachment figure as nearby, accessible, and attentive may be more likely to experience insecure attachment bond. Given that the existence of a childhood trauma may affect the attachment bond, the association between childhood trauma and AUD severity could be explained by a higher risk of insecure attachment (especially fearful attachment), which has been demonstrated to be strongly related to PTSD symptoms [22]. Studies investigating the potential differences in the course of PTSD between AUD patients with and without childhood trauma could improve our ability to tailor evidence-based interventions for AUD patients.

In this study, we sought to identify the factors associated with poorer PTSD outcomes among AUD patients hospitalized in an addiction rehabilitation center, who have a more severe form of AUD and a higher prevalence of PTSD [23]. In addition, it is possible to screen for and treat PTSD among hospitalized patients and to observe the course of PTSD over the medium term, thus facilitating the study of predictors of PTSD outcomes. Although PTSD is highly comorbid with AUD in hospitalized patients, to the best of our knowledge there are no longitudinal data about the variables associated with the course of PTSD in this specific population.

The main objective of this study was to determine how many patients hospitalized for an AUD and with a comorbid PTSD remitted from their PTSD at the end of their hospitalization, and to identify the risk factors for non-remission (i.e., socio-demographic characteristics, baseline AUD or PTSD severity, and existence of traumatic life events, including those that occurred in childhood). We hypothesized that remission rates would be lower in patients with a history of childhood trauma, and that there would be no association with baseline AUD severity, PTSD severity, other types of trauma experienced, nor with age, gender or marital status.

## 2. Experimental Section

### 2.1. Participants and Procedure

We recruited all consecutive AUD patients admitted to the “Le Courbat” addiction rehabilitation center (Centre–Val de Loire region, France) between January 2016 and October 2017. “Le Courbat” is a national referral center for the treatment of people with AUD. In the last 10 years, it has been developing specific programs for patients with a co-occurring PTSD and AUD.

The study flow chart is presented in Figure 1. Patients were considered eligible for the study if they were aged at least 18 and if they were hospitalized for an AUD as diagnosed by an addiction specialist met at baseline (*n* = 356). Eligible patients were then proposed to participate to the study and we asked them to provide their informed and signed consent if they agreed (information was given by the person in charge of the data collection (P.A.) that the participation was free and that their decision would not modify their treatment protocol during their hospitalization). Out of these 356 patients, 53 refused to participate. Patients were then asked to complete self-administered questionnaires one week after admission (T1 = baseline) using digital tablets or computers provided specifically for this study and with the help of P.A. if they had difficulties in understanding the questions. The questionnaires were designed and completed online using Sphinx mobile iQ 2 software during a systematic visit with the person in charge of the data collection (P.A.). Out of these 303 patients, 298 had fully exploitable questionnaires at T1 (five patients had missing data for at least one questionnaire including the AUDIT (Alcohol Use Disorders Identification Test), PCL-5 (PTSD Checklist for DSM-5, i.e., Diagnostic and Statistical Manual for Mental Disorders, 5^th^ edition), LEC-5 (Life Events Checklist for DSM-5), and CTQ-SF (Childhood Trauma Questionnaire, Short-Form); there was no significant difference between these five patients and the 298 others in terms of age, gender, AUDIT total score, number of traumatic events experiences, or CTQ sub-scores), including 149 patients who had a PTSD according to the LEC-5 and the PCL-5 (see the Measures subsection for the PTSD diagnostic criteria). Patients were then asked to complete again these self-administered questionnaires eight weeks after admission (T2; one week before discharge) using the same digital tablets or computers provided specifically for this study; this was proposed during a systematic visit conducted by the person in charge of the study collection (P.A.). We chose this eight-week period to match the length of stay in this rehabilitation center. Our final sample (*n* = 91; attrition rate was 38.9%) was composed of inpatients diagnosed with a co-occurring AUD and PTSD at baseline and who completed the self-administered questionnaires in full at both T1 and T2. Patients with PTSD who were included in this study (*n* = 91) did not differ from patients with PTSD who were lost to follow-up at T2 (*n* = 58) in terms of age, AUDIT total score, nor PCL-5 total score.

All patients underwent the same treatment protocol during their hospitalization: a basic treatment protocol (i.e., systematic and regular consultations with a physician expert in addiction medicine and with a physician expert in sport medicine, as well as consultations with other health care professionals: nurse, psychologist, dietician, social worker, fitness trainer, and art therapist), and an additional PTSD module for patients who screened positive for PTSD. This PTSD module, that included group-sessions with a psychologist expert in PTSD, consisted of psycho-education and information about PTSD. There was no difference in terms of treatment protocol between patients who remitted from PTSD versus patients who did not remit from PTSD.

### 2.2. Measures

For each patient, we collected data regarding socio-demographic characteristics (age, gender, and marital status), PTSD, AUD severity, and different types of traumatic events, including those that occurred in childhood.

#### 2.2.1. Childhood Trauma

History of childhood trauma was assessed at T1 using the Childhood Trauma Questionnaire, Short-Form (CTQ-SF; Bernstein et al., 2003 [24]; French validation by Paquette et al., 2004 [25]). The CTQ is a 28-item screening tool for a history of maltreatment during childhood. It measures five types of maltreatment: physical abuse (cut-off score ≥ 11), emotional abuse (cut-off score ≥ 16), sexual abuse (cut-off score ≥ 11), physical neglect (cut-off score ≥ 14), and emotional neglect (cut-off score ≥ 18). Participants answer items on a 5-point Likert scale, with responses ranging from “never true” to “very often true”. The internal reliability of the CTQ-SF was excellent for the total score (α = 0.95), good to excellent for four dimensions (Cronbach’s alphas respectively 0.81–0.86, 0.84–0.89, 0.92–0.95, and 0.88–0.91), and acceptable for physical neglect (Cronbach’s alpha ranging from 0.61–0.78) [24]. The 5-factor solution proved to be invariant across disordered-control comparison groups [26]. In this study, we used the CTQ-SF sub-scores as indicators of childhood trauma severity.

#### 2.2.2. Lifetime Exposure to Traumatic Events

We assessed the history of traumatic events (lifetime exposure to 17 types of trauma) using the Life Event Checklist for DSM-5 (LEC-5; original version: Weathers et al., 2013 [27]; French adaptation by Montreal trauma study center, 2015). This tool screens for 17 potentially traumatic events in the respondent’s lifetime, clustered in seven types: (1) natural disasters; (2) accidents; (3) physical aggressions; (4) sexual aggressions; (5) war-related trauma; (6) exposure to illness, injury, or death experiences; and (7) exposure to any other very stressful event or experience. In this study, in line with previous research, we combined these types of traumatic events into the following categories: (1) natural disasters or accidents; (2) physical or sexual aggression; (3) war-related trauma; (4) exposure to illness, injury, or death experiences; and (5) exposure to any other very stressful event or experience. The LEC-5 is often used in combination with other measures, such as the PCL-5, for the purpose of establishing exposure to a traumatic event corresponding to DSM-5 criterion A [27].

#### 2.2.3. Post-Traumatic Stress Disorder

We assessed PTSD symptoms and severity at T1 and T2 with the PTSD Checklist for DSM-5 (PCL-5; Blevins et al., 2015 [28]; French validation: Ashbaugh et al., 2016 [29]). This 20-item self-administered questionnaire assesses PTSD symptoms using a Likert-type scale for each symptom, which can be divided into four sub-scales, with scores ranging from 0 (not at all) to 4 (extremely). It also assesses severity of symptoms (score 0 to 80; cut-off ≥ 31), with sub-scores for re-experiencing (0 to 20), avoidance (0 to 8), negative alteration in cognition and mood (0 to 24), and arousal (0 to 28) [28]. We asked the patients to refer to their worst traumatic event when completing the PCL-5.

In line with the DSM-5 diagnostic criteria, PTSD was diagnosed when participants had experienced at least one traumatic event (criterion A, as assessed by the LEC-5), indicated one or more of the intrusion symptoms (criterion B), one symptom of persistent avoidance of stimuli associated with the traumatic event (criterion C), two symptoms of negative alterations in cognitions and mood (criterion D), and two symptoms of marked alterations in arousal and reactivity (criterion E). We also assessed PTSD severity using the total PCL-5 score. PTSD was considered to be in remission when the PCL-5 score decreased by 30% or more between T1 and T2 [30,31,32].

#### 2.2.4. Alcohol Use Disorder

In addition to the AUD diagnosis made by clinical assessment at baseline (T1), we assessed its severity at T1 and T2 using the Alcohol Use Disorders Identification Test (AUDIT; original version: Saunders et al., 1993 [33]; French validation: Gache et al., 2005 [34]). The AUDIT was developed in collaboration with the World Health Organization (WHO) and includes 10 questions about level of consumption, symptoms of dependence and alcohol-related consequences (cut-off ≥ 8). Its internal consistency was found to be high and test–retest data suggest good reliability (α = 0.86) and sensitivity of 0.90 [33]. We used the AUDIT total score to assess AUD severity.

### 2.3. Statistical Analyses and Ethics

Analyses were conducted using SPSS^®^ version 22 (IBM Corp. Released 2013. IBM SPSS Statistics for Windows, Version 22.0., IBM Corporation, Armonk, NY, USA). All analyses were two-tailed; *p*-values ≤ 0.05 were considered statistically significant.

Descriptive statistics included percentages for ordinal variables and means and standard deviations for continuous variables. We analyzed the correlations between our variables (Spearman’s correlation tests). First, we determined the variables associated with PTSD status (remitted vs. non-remitted) using univariate analyses: either mean comparison tests (Mann–Whitney U test, when the distribution did not follow a normal distribution, with the corresponding z-value or Student’s test with the corresponding t-value) or chi-squared tests (Pearson’s chi-squared test or Fisher’s exact test if expected frequencies were <5 in at least one cell), depending on the type of variable studied. We then used multivariate analyses (multiple linear regressions) to determine whether PTSD characteristics (each PCL-5 cluster), type of trauma encountered (including a history of childhood trauma) were significant predictors of remitted PTSD after adjusting for age (*p* was <0.20 in univariate analysis). For each dependent variable, we specified the beta regression coefficient, its 95% confidence interval, and its associated t-value and *p*-value.

This study obtained the approval of an institutional review board (Comité d’Éthique pour les Recherches Non Interventionnelles [CERNI] Tours-Poitiers) in July 2015, prior to the beginning of the study. All collected data were in line with the French recommendation regarding use of personal data, with the approval of the French Commission Nationale de l’Informatique et des Libertés (CNIL).

## 3. Results

### 3.1. Descriptive Statistics

Descriptive statistics for socio-demographic characteristics, AUD severity, PTSD severity, and different types of traumatic event (including those that occurred in childhood) at T1 are presented in Table 1. At baseline (T1), the majority of participants were male (82.4%), with a mean age of 42.8 ± 8.6 years; mean AUDIT score was 27.8 ± 7.7 (100% had an AUDIT score ≥8), and PCL-5 total score was 46.4 ± 10.9 (92.3% had a PCL-5 score ≥31). The most common categories of trauma were, in order of prevalence: accidents or natural disasters (87.9%; 82.4% experienced any accident and 46.2% experienced any natural disaster); physical or sexual aggression (82.4%; 82.4% experienced any physical aggression and 23.1% experienced any sexual aggression); any other very stressful event or experience (75.8%); exposure to illness, injury, or death experience (30.8%); and war-related trauma (6.6%). The most common categories of childhood traumatic experience were, in order of prevalence (as defined by a CTQ sub-score higher or equal to the corresponding cut-off): emotional abuse (33%); emotional negligence (29.7%); physical abuse (20.9%); physical negligence (13.2%); and sexual abuse (12.1%). Based on the LEC-5 and the CTQ, all patients experienced at least two traumatic experiences, i.e., all patients had multiple traumas. Based on the PCL-5 cutoffs, PTSD remission between T1 and T2 was observed in 74.7% of the sample. Mean total PCL-5 score dropped between T1 and T2 from 46.4 ± 10.9 to 22.6 ± 15.6 (*p* < 0.001).

### 3.2. Factors Associated with PTSD Remission in Univariate Analyses

Table 2 presents the baseline characteristics at T1 associated with PTSD remission at T2. Patients with non-remitted and remitted PTSD did not differ in terms of age, gender, or marital status. PTSD remission was not associated with the severity of AUD or PTSD at baseline. There was no difference between patients with and without remitted PTSD in terms of type of trauma encountered during lifetime (LEC-5). The only factors associated with lower remission rates were physical and emotional abuse during childhood, as assessed by the CTQ-SF.

### 3.3. Factors Associated with PTSD Remission in Multivariate Analyses

Table 3 presents the multiple logistic regression model showing the baseline characteristics at T1 associated with PTSD remission at T2. After adjustment for age, the factors significantly associated with PTSD remission at T2 were physical abuse, emotional abuse, sexual abuse, and physical negligence during childhood, but not AUD severity, nor the other types of trauma experienced during lifetime (i.e., catastrophe, accident, physical assault, sexual assault, death, war, or any other kind of trauma). PTSD remission was not associated with baseline socio-demographic characteristics.

## 4. Discussion

The main objective of this study was to determine the baseline variables associated with PTSD remission among patients hospitalized for an AUD and with a comorbid PTSD. One of the key findings is that remission varied according to the existence of trauma during a particularly important period in one’s life (i.e., childhood), but not in relation to baseline AUD severity, PTSD severity, or exposure to other types of traumatic event during lifetime.

Our results thus demonstrate that childhood traumas are predictors of non-remitted PTSD. First, our study confirms the high rate of childhood traumas in patients with AUD and PTSD, compared to all other types of trauma. In a study of AUD patients in an addiction rehabilitation center, Huang et al. found that the prevalence of childhood trauma was 55.1%, physical abuse 31.1%, emotional abuse 21.4%, sexual abuse 24%, physical neglect 19.9%, and emotional neglect 20.4% [18]. The relatively lower prevalence of sexual abuse observed in our study can be explained by the overrepresentation of men, who are less exposed to sexual abuse than women [35].

We also found that all types of childhood abuse (i.e., physical, emotional, and sexual) were associated with PTSD. In line with Bowlby’s theoretical framework and attachment theory [21], this association could be explained by a higher risk of insecure attachment (especially fearful attachment), which has been demonstrated to be strongly related to PTSD symptoms [22]. Childhood trauma may also increase the risk of some high-risk personality traits (i.e., neuroticism or low extraversion) [36], and of psychiatric disorders such as bipolar disorder, anxiety disorders or treatment-resistant depression [18,37,38,39]. It could also lead to the possibility of more severe, earlier, and long-lasting psychiatric disorders, and thus higher risk for PTSD. Finally, early trauma may increase stress vulnerability through gene–environment interactions and impaired hypothalamic–pituitary–adrenal axis response [38]. A recent study by O’Hare et al. highlighted the increased risk for vasovagal syncope in patients with childhood trauma, which may be due to paradoxical parasympathetic overdrive in response to a sympathetic increase in heart rate, with a stress response being decoupled from the original acute stressor [40].

Contrary to the findings of a meta-analysis that some types of trauma were associated with better remission rates (i.e., natural disasters were associated with 60% remission rate vs. 31.4% when the trauma was related to physical health) [41], we found no evidence of association between the type of trauma and the course of PTSD. This could be explained by the fact that the patients in our sample had been exposed to more than one type of traumatic event, making it difficult to disentangle the effects of each specific trauma on PTSD outcome. Although PTSD symptoms at baseline were associated with AUD severity at baseline, the course of PTSD was surprisingly not associated with the baseline intensity of AUD or PTSD. The lack of association with AUD severity is in line with findings that improvements in PTSD had a greater positive impact on alcohol dependence symptoms than the reciprocal relationship [19]. The association between PTSD and AUD severity at baseline is in line with previous studies [42,43], and can be explained by the amnestic, anxiolytic, and sedative properties of alcohol that may help these patients to cope with and avoid the intrusive PTSD symptoms (self-medication hypothesis) [44]. On the other hand, the lack of association between PTSD outcome and at baseline might suggest that it is not the intensity of PTSD itself that affects outcome, but rather the fact that the trauma was experienced during a particularly important period in one’s life (i.e., a trauma experienced during childhood could lead to earlier changes in personality traits or psychiatric disorders, and be associated with a long-established PTSD that could be harder to treat than a more recent one).

Another interesting finding of our study is the significant drop in the PTSD symptom score between baseline and eight weeks later. To our knowledge, this has not been assessed in previous longitudinal studies of patients with AUD and PTSD in an addiction rehabilitation center. A meta-analysis conducted by Morina et al., which focused on the course of PTSD and was not limited to AUD patients, found a spontaneous PTSD remission rate (without any specific treatment) of approximately 44% at 40-months follow-up [41]. Patients with co-occurring AUD and PTSD are an at-risk population, for whom inpatient treatment may be an interesting option when they do not respond to outpatient treatment. Addiction rehabilitation centers provide multidisciplinary care, offering patients new coping strategies, with beneficial effects on eating and sleeping habits, and reducing stress related to work or family. However, we cannot rule out the possibility that the positive PTSD outcome we observed was due to improvement in AUD, through the beneficial effects of prolonged withdrawal, or due to confounding factors associated with hospitalization (i.e., increased social support, improved physical activity, treatment of concurrent medical conditions). In addition, the study design (no control group) precludes us from demonstrating a beneficial effect of an inpatient rehabilitation program on PTSD. To demonstrate the potential beneficial effects of such programs, future studies should compare the evolution of inpatients with versus without a rehabilitation program (control-group).

At a practical level, our study highlights the importance of identifying those patients with AUD and PTSD who have a history of childhood trauma as a potential at-risk population requiring tailored treatment. These patients may have specific needs and expectations about treatment, given their psychological and psychiatric profile (more prevalent insecure attachment, more severe and earlier psychiatric disorders). Integrative psychological and psychosocial interventions focusing on trauma-related symptoms and alcohol dependence tailored to individual needs offer an interesting way to manage this vulnerable population and improve treatment outcomes [19]. One clinical implication of our study is that hospitalized AUD patients should be systematically screened for childhood abuse or neglect. Secondly, they should be assessed using different biological and psychological theoretical frameworks in order to better understand how these traumas may be linked to the co-occurring addictive disorder and PTSD. Our results also highlight the need for close and long-term follow-up of this at-risk population.

This study has several limitations. First, PTSD diagnosis and AUD severity were based on self-administered questionnaires, which increases the risk of false positives and possible recall bias. Given that the LEC-5 assesses lifetime exposure to trauma and the CTQ-SF assesses childhood trauma, there may be an overlap between the two measures. Although we knew if a trauma occurred during childhood, we did not assess the exact timing of the trauma during childhood or adulthood. Another limitation concerns the participants in our sample (AUD inpatients in a rehabilitation center), who may differ from other patients with AUD and comorbid PTSD (inpatients usually exhibit more severe AUD/PTSD). This selection bias and our small sample of patients with non-remitted PTSD may limit the generalizability of our findings. Finally, our study was conducted in a single center, where the other psychosocial interventions proposed concurrently may have impacted the course of PTSD.

## 5. Conclusions

We demonstrated that the main factor associated with poor PTSD outcome in patients with comorbid AUD and PTSD was a history of childhood trauma. We also found that neither the type of trauma experienced, nor the initial severity of AUD were associated with PTSD remission. These results highlight the importance of systematically assessing the history of childhood trauma in patients hospitalized for an AUD, and of tailoring evidence-based interventions for this high-risk population. Future studies should test the efficacy of trauma-focused interventions (i.e., EMDR or Cognitive and Behavioural Therapy) for treating both PTSD and AUD, and compare its efficacy in patients with and without a history of childhood trauma [45,46]. Tailored interventions for patients with AUD and PTSD could help us to meet the challenge of improving the therapeutic management of patients with these dual disorders [4].

## Figures and Tables

**Figure 1 jcm-09-02054-f001:**
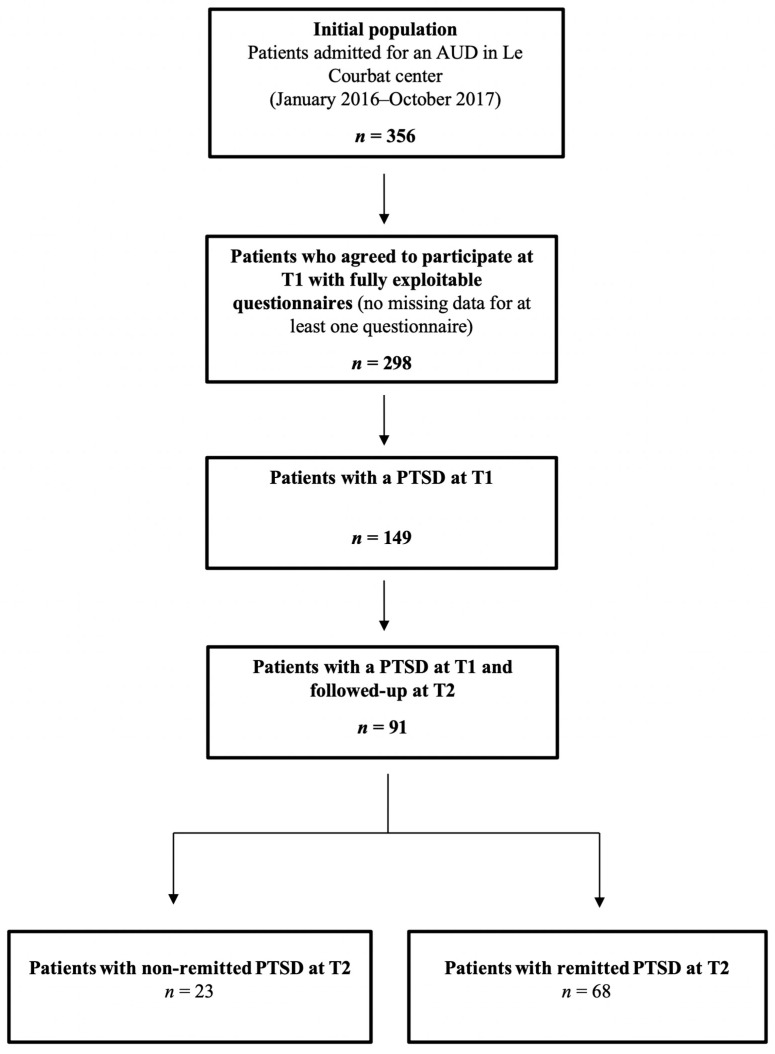
Study flow chart. Note: AUD, Alcohol Use Disorder; AUDIT, Alcohol Use Disorders Identification Test; PTSD, Post-Traumatic Stress Disorder. T1, Assessment one week after admission (baseline assessment); T2, Assessment eight weeks after initial assessment (one week before discharge).

**Table 1 jcm-09-02054-t001:** Descriptive statistics of the sample at baseline (T1; *n* = 91).

	Mean ± SD or Percentage (number)
Socio-Demographic Characteristics at T1	
Age (years)	42.9 ± 8.6
Gender (male)	82.4% (85)
Marital status (married or in a relationship)	36.3% (33)
Alcohol use disorder severity at T1 (AUDIT total score)	27.8 ± 7.7
PTSD severity at T1 (PCL-5 total score)	46.4 ± 10.9
Lifetime exposure to a traumatic event (LEC-5)	
At least one traumatic event	100% (91)
Natural disaster or/and accident (at least one)	87.9% (80)
Physical or/and sexual aggression (at least one)	82.4% (75)
War-related trauma (at least one)	6.6% (6)
Illness, injury, or death experience (at least one)	30.8% (28)
Any other traumatic event (at least one)	75.8% (69)
Childhood trauma (CTQ sub-scores)	
CTQ physical abuse score	8.3 ± 4.8
CTQ emotional abuse score	12.9 ± 5.9
CTQ sexual abuse score	6.8 ± 3.9
CTQ physical negligence score	8.9 ± 3.7
CTQ emotional negligence score	12.8 ± 5.0

Note: Descriptive data are presented as mean ± standard deviation (SD) or percentage (number). AUDIT, Alcohol Use Disorders Identification Test; CTQ, Childhood Trauma Questionnaire, short form; LEC-5, Life Event Checklist for DSM-5; PCL, PTSD Checklist for DSM-5; PTSD, Post-traumatic Stress Disorder; T1, Assessment at baseline (i.e., one week after admission).

**Table 2 jcm-09-02054-t002:** Baseline characteristics at T1 associated with PTSD remission at T2 in univariate analyses (*n* = 91).

	Non-Remitted PTSD at T2 (*n* = 23)	Remitted PTSD at T2 (*n* = 68)	Statistic Test
Socio-demographic characteristics at T1			
Age (years)	40.0 ± 7.9	43.7 ± 8.7	t = −1.83
Gender (male)	73.9% (17)	85.3% (58)	χ^2^ = 1.54
Marital status (married or in a relationship)	26.1% (6)	39.7% (27)	χ^2^ = 1.38
Alcohol use disorder severity at T1 (AUDIT total score)	27.0 ± 7.3	28.0 ± 7.9	t = −0.56
PTSD severity at T1 (PCL-5 total score)	43.8 ± 8.8	47.3 ± 11.4	t = 1.34
Lifetime exposure to a traumatic event (LEC-5)			
Natural disaster or/and accident (at least one)	91.3% (21)	86.8% (59)	χ^2^ = 0.33
Physical or/and sexual aggression (at least one)	78.3% (18)	83.8% (57)	χ^2^ = 0.37
War-related trauma (at least one)	8.6% (2)	5.9% (4)	χ^2^ = 0.22
Illness, injury, or death experience (at least one)	39.1% (9)	29.4% (20)	χ^2^ = 0.75
Any other traumatic event (at least one)	78.3% (18)	75% (51)	χ^2^ = 0.10
Childhood trauma (CTQ sub-scores)			
CTQ physical abuse score *	10.7 ± 6.5	7.5 ± 3.8	t = 2.22
CTQ emotional abuse score *	15.1 ± 6.2	12.1 ± 5.6	t = 2.16
CTQ sexual abuse score	8.4 ± 6.1	6.2 ± 2.7	t = 1.64
CTQ physical negligence score	10.3 ± 3.8	8.4 ± 3.2	t = 1.75
CTQ emotional negligence score	13.7 ± 6.1	12.5 ± 4.6	t = 0.99

Note: Descriptive data are presented as mean ± standard deviation or percentage (number). We compared patients with remitted vs. non-remitted PTSD using mean comparison tests (Mann–Whitney U test or Student’s test) and chi-squared tests. * indicates the variables significantly associated with PTSD remission. AUDIT, Alcohol Use Disorders Identification Test; CTQ, Childhood Trauma Questionnaire, short form; LEC-5, Life Event Checklist for DSM-5; PCL, PTSD Checklist for DSM-5; PTSD, Post-traumatic Stress Disorder; T1, Assessment at baseline (i.e., one week after admission); T2, Assessment eight weeks after admission (i.e., one week before discharge).

**Table 3 jcm-09-02054-t003:** Baseline T1 characteristics associated with PTSD remission at T2 in multiple logistic regressions after adjustment for age.

	Non-Remitted PTSD	Remitted PTSD	Chi-Squared	*p*-value	Odds-Ratio	CI 95%
	at T2 (*n* = 23)	at T2 (*n* = 68)				
Socio-demographic characteristics						
Gender (male)	73.9% (17)	85.3% (58)	2.05	0.15	0.42	0.13–1.38
Marital status (married or in a relationship)	26.1% (6)	39.7% (27)	1.57	0.21	0.5	0.17–1.47
Alcohol use disorder severity at T1 (AUDIT total score)	27.0 ± 7.3	28.0 ± 7.9	0.46	0.5	1.02	0.96–1.09
PTSD severity at T1 (PCL-5 total score)	43.8 ± 8.8	47.3 ± 11.4	1.94	0.16	1.03	0.99–1.08
Life events (LEC-5)						
Natural disaster or/and accident (at least one)	91.3% (21)	86.8% (59)	0.27	0.61	0.1.54	0.30–7.92
Physical or/and sexual aggression (at least one)	78.3% (18)	83.8% (57)	0.49	0.49	0.65	0.19–2.18
War-related trauma	8.6% (2)	5.9% (4)	1.2	0.27	2.98	0.42–20.99
Illness, injury, or death experience	39.1% (9)	29.4% (20)	1.77	0.18	2.05	0.71–5.91
Any other traumatic event	78.3% (18)	75% (51)	0.13	0.72	1.24	0.23–3.02
Childhood trauma (CTQ sub-scores)						
CTQ physical abuse *	10.7 ± 6.5	7.5 ± 3.8	7.4	<0.001 *	0.8741	0.79–0.96
CTQ emotional abuse *	15.1 ± 6.2	12.1 ± 5.6	4.8	0.03 *	0.91	0.84–0.99
CTQ sexual abuse *	8.4 ± 6.1	6.2 ± 2.7	4.8	0.03 *	0.88	0.78–0.99
CTQ physical negligence *	10.3 ± 3.8	8.4 ± 3.2	4.05	0.05 *	0.88	0.78–0.99
CTQ emotional negligence	13.7 ± 6.1	12.5 ± 4.6	1.98	0.16	0.93	0.39–3.90

Note: Descriptive data are presented as mean ± standard deviation or percentage (number). We compared patients with remitted vs. non-remitted PTSD using multiple logistic regression adjusted for age. * indicates the variables significantly associated with PTSD remission. AUDIT, Alcohol Use Disorders Identification Test; CTQ, Childhood Trauma Questionnaire, short form; LEC-5, Life Event Checklist for DSM-5; PCL, PTSD Checklist for DSM-5; PTSD, Post-traumatic Stress Disorder; T1, Assessment at baseline (i.e., one week after admission); T2, Assessment eight weeks after admission (i.e., one week before discharge).

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
