# Peer review of "Childhood Trauma Predicts Less Remission from PTSD among Patients with Co-Occurring Alcohol Use Disorder and PTSD"

_jcm, 2020, doi:10.3390/jcm9072054_

Round 1

Reviewer 1 Report

PTSD is associated, with alcohol use disorder (AUD). People who have AUD have higher rates of PTSD than the general population. For these clients with dual problems, clinical outcomes are poorer. The purpose of this study is to identify factors associated with poorer PTSD outcomes in AUD clients hospitalized in an addiction rehabilitation center. The au sought to examine patients that remitted from PTSD at the end of hospitalization and identify risk factors for non-remission. The topic is an important one due to the prevalence of co-occurring AUD and PTSD and the tenacity of both disorders.

Participants were recruited from all patients referred to an addiction rehabilitation center for a 19 month period. Participants completed an informed consent and it is a strength that the study was approved by an institutional review board.

There is no indication of who recruited, how participants were approached, how the researchers guarded against feelings of participant coercion, who assisted or monitored participant completion of measures, how the researchers were sure the participants could read the measures. These issues should be addressed.

The number of people who were recruited, the number that said yes, and the number that was screened in should be reported earlier in the procedures and long before a flow chart near the end of the paper. I was looking for those data earlier.

The 91 participants were those who completed the questionnaires at T1 and T2 fully and had PTSD. Attrition rates appear to be zero but it’s not really clear because clients had to complete T1 and T2 fully to be in the study. This should be clarified.

Measures

Measures used were highly regarded measures.

The study is characterized as longitudinal but the follow up assessment is only 8 weeks. A rationale should be given for this time period as well as how participants could be fully treated for PTSD in this short time frame. It would be helpful to know if the nonremission participants had equal treatment to the remission. The lack of treatment or lower number of sessions might account for the nonremission.

Severity of childhood trauma or the experience of multiple traumas was not reported.

Au should include information showing how long ago the index trauma occurred and if the index trauma was the child maltreatment or something in adulthood.

It would be helpful to know the developmental period in which the participant experienced the childhood trauma.

Results

The n’s of the two groups are vastly different. At the end of the day, the study that sets out to look at factors related to nonremission is only dealing with 23 participants. This small n could produce spurious findings and could impact generalizability.

In the results section it would be helpful to include clinical cutoffs for the AUDIT and PCL so the reader can understand what the scores mean.

In the discussion, page 4 of 16, the au states that the study demonstrates the beneficial effects of rehabilitation programs for dual diagnosis. This is an overstatement of the data. The participants may have fared equally as well or better in an outpatient program or no treatment. This is where a comparison group would be necessary.

There is no presentation of the treatment that participants experienced. It should be noted that the scientific literature is clear that inpatient care is not associated with better outcomes than outpatient. In addition, it was not clear what treatment was being used for the PTSD but part of evidence-based treatment is in vivo exposure of feared triggers and this cannot be conducted in a hospital setting. It would be helpful to understanding what they received.

The point about screening patients for childhood trauma is well taken and truly should be standard practice.

Author Response

Itemized list of reviewers comments with corresponding revisions or responses

Manuscript ID: JCM-809732

Decision: Major revision

Reviewer#1 Comments

Comment#1. There is no indication of who recruited, how participants were approached, how the researchers guarded against feelings of participant coercion, who assisted or monitored participant completion of measures, how the researchers were sure the participants could read the measures. These issues should be addressed.

Reply: We agree that these important points were not detailed in our Methods section. This study is part of a larger protocol that was initiated in January following a collaboration between the Courbat Team and the Department of Psychiatry and Addiction of the University Hospital of Tours. This collaboration was built in response to a strong demand of this rehabilitation center to better address and treat PTSD in patients hospitalized in the rehabilitation center (this rehabilitation center has a nation-wide recruitment for patients with PTSD). The study was systematically proposed to every patient admitted in the rehabilitation center during a systematic visit at baseline with the person in charge of the study collection (PA). All hospitalized patients were proposed to participate to the study and we asked them to provide their informed and signed consent if they agreed (information was given by the person in charge of the data collection [PA] that the participation was free and that their decision would not modify their treatment protocol during their hospitalization). This study was in line with the ethical standards of scientific research, as indicated by the approval of an institutional review board prior to the beginning of the study. If patients agreed to participate, the questionnaires were then completed by patients during this visit using computer-assisted measures, with the help of the person in charge of the data collection (PA) if they encountered some difficulties in understanding or answering the questions. Patients were asked to complete questionnaires 8 weeks later (T2). These facts may explain why the participation rate is high.

In line with this comment and with Reviewer#1 Comment#2, we have added these important points in the Methods section, Participants and procedure subsection in a paragraph (page 3 and page 4, line 133 to 164), and we revised the study flow chart presented in Figure 1 to indicate the exact number of patients at baseline and the number of patients excluded as well as the motives for exclusion (page 5; see also answer to reviewer#1 Comment#2 above).

Comment#2. The number of people who were recruited, the number that said yes, and the number that was screened in should be reported earlier in the procedures and long before a flow chart near the end of the paper. I was looking for those data earlier. The 91 participants were those who completed the questionnaires at T1 and T2 fully and had PTSD. Attrition rates appear to be zero but it’s not really clear because clients had to complete T1 and T2 fully to be in the study. This should be clarified.

Reply: Thank you for this comment, which is line with the Reviewer#1 Comment#1. We clarified this point, as indicated in the Methods section (page 3 and page 4, line 131 to 165) as well as in the revised study flow chart (page 5).

Comment#3. The study is characterized as longitudinal but the follow up assessment is only 8 weeks. A rationale should be given for this time period as well as how participants could be fully treated for PTSD in this short time frame. It would be helpful to know if the nonremission participants had equal treatment to the remission. The lack of treatment or lower number of sessions might account for the nonremission.

Reply: We choose this (short) time period of eight weeks because this rehabilitation center proposes hospitalization with a therapeutic protocol that lasts approximately 10 weeks. This timeline enabled us to assess patients during their hospitalization, thus limiting the risk of loss to follow-up and the potential impact of confounding factors (e.g., events that would have occurred outside the hospitalization and that would have been different between patients). This is now indicated in the manuscript (page 4 line 163 to 164).

The second part of the comment is in line with Reviewer#1 Comment#9. In our first draft, there was indeed no indication of the type of treatments received by the patients. In this addiction rehabilitation center, all patients underwent the same basic treatment protocol, with an additional PTSD module for patients who screened positive for PTSD (i.e., all the patients from our study) (informations added in the manuscript on page 4 line 169 to 175):

  • The basic treatment protocol included a systematic consultation with a physician expert in addiction medicine and a systematic consultation with a physician expert in sports medicine at baseline (during the first week after admission). These two consultations were renewed every 2 weeks until the end of the hospitalization (all patients had five consultations with a physician expert in addiction medicine and five consultations with a physician expert in sports medicine). The basic treatment protocol also included consultations with other health care professionals (including a systematic consultation with a nurse, a psychologist, a dietician, a social worker, a fitness trainer, and an art therapist). All patients underwent the same protocol in terms of consultations.
  • The additional module for PTSD was the same for all patients who screened positive for PTSD (i.e., all patients included in our study): a psychologist expert in PTSD conducted group sessions with psycho-education and information about PTSD (same number of group sessions for all patients). We did not conduct in vivo exposure of feared triggers during their hospitalization.

            Thus, patients who remitted from their PTSD had the same treatment protocol than patients who did not remit from their PTSD (i.e., same number of individual and group sessions). This is now indicated in the manuscript (page 4 line 175 to 176).

Comment#4. Severity of childhood trauma or the experience of multiple traumas was not reported.

Reply: The experience of multiple trauma was not reported in our first manuscript. Out of the 91 patients with PTSD, no patient experienced a single trauma : all patients experienced at least two traumatic event as assessed by the CTQ and the LEC-5. This point has been added to the manuscript (page 8 line 317 to 318).

The severity of childhood trauma (CTQ subscores) are presented in Table 1 (page 9) and the percentages of patients having CTQ subscores over the cut-offs are presented in the results section (page 8, line 314 to 317).

Comment#5. Au should include information showing how long ago the index trauma occurred and if the index trauma was the child maltreatment or something in adulthood. It would be helpful to know the developmental period in which the participant experienced the childhood trauma.

Reply: This comment is in line with Reviewer#2 Comment#5. In this study, we use two validated measures of trauma: we assessed a history of childhood trauma using the CTQ, and the lifetime existence of traumatic event using the LEC-5. These measures enabled us to determine whether the patient had a history of childhood trauma, and whether he encountered traumatic event during his lifetime. However, the CTQ does not assess the exact timing of the trauma during childhood, and the LEC-5 does not assess the exact timing of the traumatic event. We therefore added this point as a limitation to our present study (page 14 line 605 to 608).

Comment#6. Results. The n’s of the two groups are vastly different. At the end of the day, the study that sets out to look at factors related to nonremission is only dealing with 23 participants. This small n could produce spurious findings and could impact generalizability.

Reply: The reviewer states here that our sample size (n=91), and more specifically, the fact that 23 of these 91 patients (=25.3%) had no PTSD remission, could impact the generalizability of our findings. In this study, our final analyses are based on the comparison of PTSD patients with (n=68) versus without remission (n=23), and we agree with the reviewer that a larger sample size would enable more robust results. However, when we compared the 91 PTSD patients who participated in the study (patients with PTSD at T1 with complete follow-up at T2) versus the 58 PTSD patients who did not participate in the study (patients with PTSD at T1 but with loss to follow-up at T2), we found no significant differences in terms of age (p=.57), AUDIT total score (p=.17), nor PTSD severity score (p=.17). These findings support the hypothesis that the population of PTSD patients who participated at T1 and T2 may be comparable to the population of PTSD patients who participated at T1 but who were lost to follow-up. We added this point in our manuscript (page 4 line 175 to 176) as well as the reviewer’s comment as a limitation to our study (page 14 line 610 to 611).

Comment#7. In the results section it would be helpful to include clinical cutoffs for the AUDIT and PCL so the reader can understand what the scores mean.

Reply: Thank you for this point. We first diagnosed AUD using a clinical interview, and we then used the AUDIT score to estimate the AUD severity. As stated in Saunders et al. (1993) AUDIT validation paper, the recommended cut-off for the AUDIT is a total score ≥ 8 (“at risk for hazardous drinking”). All patients had an AUDIT score ≥ 8. For the PCL-5, we diagnosed PTSD in line with the DSM-5 diagnostic criteria: participants had experienced at least one traumatic event (criterion A, as assessed by the LEC-5), indicated one or more of the intrusion symptoms (criterion B, as assessed by the PCL-5), one symptom of persistent avoidance of stimuli associated with the traumatic event (criterion C, as assessed by the PCL-5), two symptoms of negative alterations in cognitions and mood (criterion D, as assessed by the PCL-5), and two symptoms of marked alterations in arousal and reactivity (criterion E, as assessed by the PCL-5). We then used the PCL-5 total score to assess PTSD severity. In line with Ashbaugh et al. study (2016), we can refer to a cut-off ≥31 to indicate a significant score. In our study, 92.3% had a PCL-5 score ≥ 31. We have added these cutoffs for both the AUDIT and the PCL-5 in the methods (page 7 line 272 and page 6 line 251) and in the results section (page 8 line 307 and 308).

Comment#8. In the discussion, page 4 of 16, the au states that the study demonstrates the beneficial effects of rehabilitation programs for dual diagnosis. This is an overstatement of the data. The participants may have fared equally as well or better in an outpatient program or no treatment. This is where a comparison group would be necessary.

Reply: We agree with the reviewer that this statement was overstated: the demonstration of an improvement in terms of PTSD in a follow-up study without any control group does not precludes that this improvement is due to the inpatient program, nor that the improvement would have been better than an outpatient program. We have therefore deleted this sentence from the manuscript and added it as in the limitation section of the manuscript (future studies should compare the evolution in PTSD severity between a group with a rehabilitation program and a control group without a rehabilitation program) (page 14 line 587 to 590).

In order to demonstrate the beneficial effects of an inpatient rehabilitation program, future studies should compare the evolution of inpatients with versus without a rehabilitation program.

Comment#9. There is no presentation of the treatment that participants experienced. It should be noted that the scientific literature is clear that inpatient care is not associated with better outcomes than outpatient. In addition, it was not clear what treatment was being used for the PTSD but part of evidence-based treatment is in vivo exposure of feared triggers and this cannot be conducted in a hospital setting. It would be helpful to understanding what they received.

Reply: This comment is in line with Reviewer#2 Comment#3. In our first version, there was no indication of the type of treatments received by the patients. In this addiction rehabilitation center, all patients underwent the same basic treatment protocol, with an additional PTSD module for patients who screened positive for PTSD (i.e., all the patients from our study).

            The basic treatment protocol included a systematic consultation with a physician expert in addiction medicine and a systematic consultation with a physician expert in sports medicine at baseline (during the first week after admission). These two consultations were renewed every 2 weeks until the end of the hospitalization (all patients had five consultations with a physician expert in addiction medicine and five consultations with a physician expert in sports medicine). The basic treatment protocol also included consultations with other health care professionals (including a systematic consultation with a nurse, a psychologist, a dietician, a social worker, a fitness trainer, and an art therapist). All patients underwent the same protocol in terms of consultations.

            The additional module for PTSD was the same for all patients who screened positive for PTSD (i.e., all patients included in our study): a psychologist expert in PTSD conducted group sessions with psycho-education and information about PTSD (same number of group sessions for all patients). We did not conduct in vivo exposure of feared triggers during their hospitalization.

            We added this information to the manuscript (page 4 line 169 to 175).

We would like to thank you and the reviewers for your work and for taking the time to review and comment our manuscript. We answered your comments to our best and we hope that the modifications proposed improved the manuscript’s clarity and quality.

Reviewer 2 Report

jcm-809732

Title: Childhood Trauma predicts less remission from PTSD among patients with co-occurring alcohol use disorder and PTSD

Summary: This study examines baseline predictors of non-remitted PTSD in patients hospitalized for an AUD. Ninety-one patients had comorbid PTSD and AUD and were assessed at baseline (T1) and 8 weeks later (T2) for PTSD diagnosis/severity, the type of trauma experienced, AUD diagnosis/severity and childhood trauma. Non-remission of PTSD at T2 was associated with a history of childhood trauma but not with the type of trauma experienced, nor baseline PTSD or AUD severity. This study is important as it focuses on comorbid PTSD and AUD which are frequently observed in trauma populations. Furthermore, given treatment challenges posed with this population, knowledge of factors that affect remission is important to know. However, the manuscript would be significantly improved by providing additional information and reframing some of the conclusions made.

1. Introduction: The Introduction is well written and is clear. The authors should include a description of attachment theory as an explanation for why experiencing childhood abuse can lead to poorer prognosis for PTSD, just as they did in the Discussion. In their description they should elaborate on what attachment theory is and how experiencing a traumatic event can affect one’s attachment style. Not all readers will be knowledgeable on attachment theory.

The authors note that the main objective of the study was to “determine how many patients hospitalized for an AUD and with a comorbid PTSD remitted from their PTSD at the end of the hospitalization and to identify the risk factors for non-remission (i.e., socio-demographic characteristics, baseline AUD or PTSD severity, types of trauma experienced and history of childhood traumas)”. However, ‘history of childhood trauma’ is a ‘type of trauma experienced’. As such the authors should clarify that assessment of traumatic life events, including those that occurred in childhood, will be assessed.

2. Experimental Section.

2.1. This study, while examining predictors of remission, is also a treatment outcome study. Therefore, information needs to be provided as to the type of treatment that is being implemented at rehabilitation center.

2.2. Again, the authors should separate ‘childhood trauma’ with ‘type of traumatic events’. This should be revised to something like ‘different type of traumatic events, including those that occurred in childhood’.

2.2.2. The authors used the LEC for the current study. While the LEC is commonly used to assess traumatic events, it becomes problematic because it is unclear if the events assessed in the LEC occurred during childhood. For example, if one indicates that they experienced ‘sexual aggressions’ it is unclear if they occurred during childhood or adulthood, unless the authors indicated specifically to participants that they are to identify the events only if they occurred during adulthood. Not only that, the ‘exposure to any other very stressful event or experience’ also is vague and if a participant checks that box, it is unclear what is being referenced. The authors should note that there may be overlap between the two measures (if the ‘adulthood’ specifiers was not used) and as such is a limitation in the use of this measure.

2.2.3. Were participants asked to reference their ‘worst traumatic event’ when completing the PCL-5? If so, this should be noted. If not, this should be noted as a limitation. The authors also mention that ‘PTSD was considered to be in remission when the PCL-5 score decreased by 30% or more between T1 and T2’. Please include a reference for this statement. If there is no reference, why was 30% chosen?

2.3. The authors indicate that they included the different PTSD clusters, all 17 (I believe although it is not clear) of the type of traumas encountered, and history of childhood traumas were significant predictors of remitted PTSD after adjusting for age, gender and marital status. I believe that some sort of adjustment (e.g., Bonferroni) needs to be made for the multiple comparisons made with the univariate test.

The authors might also consider combining some of the categories (particularly for the LEC) when running the multivariate analyses. This study is examining whether childhood trauma (as well as other traumas) predicts remission, not which specific traumas predict remission. Therefore, some of the LEC categories could be combined, consistent with prior research using the LEC (e.g., combine accident and natural disaster). As an initial first step, the different categories on the CTQ also could be combined and then the individual subscales examined subsequently.

3. Results. The authors mention that 8 were excluded because of ‘incomplete data’. Information on PTSD severity, trauma exposure etc. would be important to describe.

4. Discussion. The authors conclude that “remission varied according to the timing of the trauma (i.e., in childhood)”. This statement is also used in the abstract. I think that this is not entirely accurate and misrepresents what is actually being tested. The authors are not technically examining whether or not traumas that occurred earlier in time are more strongly related to decreased remission, but childhood traumas specifically. Yes, childhood traumas are technically those that occurred earlier in time, but that is not what makes them more ‘traumatic’. The authors should focus on the challenges posed when one experiences a traumatic event (particularly a sexual trauma) in childhood, during a certain developmental stage, not the fact that these events occurred early on in life. In doing so, it suggests that traumas occurred during adolescence or early adulthood would be less severe than in childhood. It’s not about the timing of the event but that these events occurred during a particularly important developmental period in one’s life.

Author Response

Thank you for your letter and the opportunity to revise our manuscript entitled “Childhood trauma predicts less remission from PTSD among patients with co-occurring alcohol use disorder and PTSD” (Manuscript ID: JCM-809732).

            We have carefully considered all of the reviewers comments, and revised our manuscript in line with these suggestions. The response to each of your comments is presented above in the Itemized list of reviewer comments with corresponding revisions or responses (please see the corresponding file attached).       
            The manuscript has been revised in line with these comments and, for better clarity, we uploaded a revised version of the manuscript (which includes changes in MS Word track change version) and an additional cleaned version of the manuscript at the end of the document (which corresponds to the manuscript with all changes accepted).
            We would like to thank you and the reviewers for your work and for taking the time to review and comment our manuscript. We hope that the modifications will meet your expectations as well as those of the reviewers. We hope that with these changes, the revised manuscript will be acceptable for publication in the Journal of Clinical Medicine. Thank you for your time and consideration,     
            Respectfully,

Paul Brunault on behalf of the authors        
University Hospital of Tours, France         
Tel: +33 2 47 47 80 43         
Fax: +33 2 47 47 84 02         
[email protected]

Itemized list of reviewers comments with corresponding revisions or responses

Manuscript ID: JCM-809732

Decision: Major revision

Reviewer#2 Comments

Comment#1. The authors should include a description of attachment theory as an explanation for why experiencing childhood abuse can lead to poorer prognosis for PTSD, just as they did in the Discussion. In their description they should elaborate on what attachment theory is and how experiencing a traumatic event can affect one’s attachment style. Not all readers will be knowledgeable on attachment theory.

Reply: We agree with these points. We included some sentences in the Introduction section to describe attachment theory and to propose it as a potential explanation underlying the association between childhood trauma, PTSD, and AUD (page 2 and page 3, line 90 to 109).

Comment#2. The authors note that the main objective of the study was to “determine how many patients hospitalized for an AUD and with a comorbid PTSD remitted from their PTSD at the end of the hospitalization and to identify the risk factors for non-remission (i.e., socio-demographic characteristics, baseline AUD or PTSD severity, types of trauma experienced and history of childhood traumas)”. However, ‘history of childhood trauma’ is a ‘type of trauma experienced’. As such the authors should clarify that assessment of traumatic life events, including those that occurred in childhood, will be assessed.

Reply: We agree with the reviewer and thank you for helping us clarifying our main objective. We changed this sentence and the text now reads: “determine how many patients hospitalized for an AUD and with a comorbid PTSD remitted from their PTSD at the end of the hospitalization and to identify the risk factors for non-remission (i.e., socio-demographic characteristics, baseline AUD or PTSD severity, as well as existence of traumatic life events, including those that occurred in childhood)” (page 3 line 121). In line with this comment, we also changed a sentence in the Methods section (page 6 line 220 to 221) and used the same wording throughout the manuscript in the Methods, Results and Discussion sections.

Comment#3. 2.1. This study, while examining predictors of remission, is also a treatment outcome study. Therefore, information needs to be provided as to the type of treatment that is being implemented at rehabilitation center.

Reply: This comment is in line with Reviewer#1 Comment#9: in our first version, there was no indication of the type of treatments received by the patients. In this addiction rehabilitation center, all patients underwent the same basic treatment protocol, with an additional PTSD module for patients who screened positive for PTSD (i.e., all the patients from our study).

            The basic treatment protocol included a systematic consultation with a physician expert in addiction medicine and a systematic consultation with a physician expert in sports medicine at baseline (during the first week after admission). These two consultations were renewed every 2 weeks until the end of the hospitalization (all patients had five consultations with a physician expert in addiction medicine and five consultations with a physician expert in sports medicine). The basic treatment protocol also included consultations with other health care professionals (including a systematic consultation with a nurse, a psychologist, a dietician, a social worker, a fitness trainer, and an art therapist). All patients underwent the same protocol in terms of consultations.

            The additional module for PTSD was the same for all patients who screened positive for PTSD (i.e., all patients included in our study): a psychologist expert in PTSD conducted group sessions with psycho-education and information about PTSD (same number of group sessions for all patients). We did not conduct in vivo exposure of feared triggers during their hospitalization.

            We added these informations in the Methods section (page 4 line 169 to 175).

Comment#4. 2.2. Again, the authors should separate ‘childhood trauma’ with ‘type of traumatic events’. This should be revised to something like ‘different type of traumatic events, including those that occurred in childhood’.

Reply: Thank you for this point, that helps us improve the clarity of the manuscript. We changed our wordings: the notion of childhood trauma is now included as a type of childhood trauma (Introduction: page 3 line 212; Methods section: page 6 line 220 to 221; we also used the same wording throughout the manuscript in the Methods, Results and Discussion sections.

Comment#5. 2.2.2. The authors used the LEC for the current study. While the LEC is commonly used to assess traumatic events, it becomes problematic because it is unclear if the events assessed in the LEC occurred during childhood. For example, if one indicates that they experienced ‘sexual aggressions’ it is unclear if they occurred during childhood or adulthood, unless the authors indicated specifically to participants that they are to identify the events only if they occurred during adulthood. Not only that, the ‘exposure to any other very stressful event or experience’ also is vague and if a participant checks that box, it is unclear what is being referenced. The authors should note that there may be overlap between the two measures (if the ‘adulthood’ specifiers was not used) and as such is a limitation in the use of this measure.

Reply: This comment is in line with Reviewer#1 Comment#5. We did not use the “adulthood” specifier for the LEC, and we agree with the reviewer that a limitation of the LEC is indeed the fact that we do not know whether the trauma mentioned in the LEC-5 occurred during childhood or later on. We agree with the fact that there may be an overlap between the LEC and the CTQ and that it is a limitation in the use of this measure and we added this point to the manuscript (page 14 line 605 to 608).

Comment#6. 2.2.3. Were participants asked to reference their ‘worst traumatic event’ when completing the PCL-5? If so, this should be noted. If not, this should be noted as a limitation.

Reply: Yes, it was indeed the case: we asked the participants to refer to their worst traumatic event when completing the PCL-5 (Instruction was the one classical used in PCL-5 studies, including “Below is a list of problems that people sometimes have in response to a very stressful experience. Keeping your worst event in mind, please read each problem carefully and then circle one of the numbers to the right to indicate how much you have been bothered by that problem in the past month”). We added this point to the manuscript (page 6 line 252 to 253).

Comment#7. The authors also mention that ‘PTSD was considered to be in remission when the PCL-5 score decreased by 30% or more between T1 and T2’. Please include a reference for this statement. If there is no reference, why was 30% chosen?

Reply: We indeed forgot to include references related to our main outcome. The improvement of at least 30% in terms if PTSD symptom score is the most commonly used definition of remission in the literature, as indicated by previous studies conducted in patients with PTSD (Zohar et al. J Clin Psychopharmacol 2002; Brady et al. JAMA 2000; Dunlop et al. Behav Science 2014). We have added these references to the manuscript (page 7 line 266).

Comment#8. 2.3. The authors indicate that they included the different PTSD clusters, all 17 (I believe although it is not clear) of the type of traumas encountered, and history of childhood traumas were significant predictors of remitted PTSD after adjusting for age, gender and marital status. I believe that some sort of adjustment (e.g., Bonferroni) needs to be made for the multiple comparisons made with the univariate test.

Reply: We indeed included all 17 types of traumas using the LEC-5; this is now stated more clearly in the manuscript (page 6 line 235). The question whether one should adjust their statistical analyses based on the number of tests used is a debated topic in the scientific literature. Some people state that the chance of finding at least one test statistically significant due to chance and incorrectly declaring a difference increases as the number of comparisons increases (i.e., classicist point of view), while some other argue that (1) p-value adjustments are calculated based on how many tests are to be considered, and that number has been defined arbitrarily and variable, and (2) p-value adjustments reduce the chance of making type I errors, but they increase the chance of making type II errors or needing to increase the sample size (i.e., rationalist point of view). In line with Feise’s approach (Feise 2002 BMC Med Res Methodol), we revised our manuscript by choosing not to adjust our univariate analyses for the number of tests used, and preferred to show our results with the magnitude of the effect rather than with the p-values. We thus deleted p-value results from Table 2 (univariate analysis) because it could be misleading (what is important is the magnitude of the effect and not the exact p-value). We revised our Table 2 accordingly (page 11).

Comment#9. The authors might also consider combining some of the categories (particularly for the LEC) when running the multivariate analyses. This study is examining whether childhood trauma (as well as other traumas) predicts remission, not which specific traumas predict remission. Therefore, some of the LEC categories could be combined, consistent with prior research using the LEC (e.g., combine accident and natural disaster). As an initial first step, the different categories on the CTQ also could be combined and then the individual subscales examined subsequently.

Reply: The reviewer suggests us to combine the LEC and the CTQ subscales before conducting multivariate analysis. As suggested by prior research using the LEC, we combined the categories “accident” and “natural disaster”, as well as the categories “physical aggression” and “sexual aggression”. We obtained results close to the original ones (no important differences in terms of effect size or p-value). The new statistical analyses are presented in Table 2 and 3 (page 11 and page 12) and we specified these combinations in the Methods section (page 6 line 240 to 243).

For the CTQ, the original work by Spinhoven et al. (2014) demonstrated that the CTQ had a 5-factor solution that proved to be invariant across disordered-control comparison groups. This demonstrates that researchers should use the five CTQ sub-scores rather than a global CTQ score, and suggests that it would be difficult to interpret results obtained with the use of a total CTQ score. We therefore kept the use of the 5 CTQ dimensions in our analyses. We however provide here the statistical analyses requested by the reviewer: when we used the CTQ as a total score, we obtained the same results; when using a logistic regression with PTSD remission as a dependent variable and CTQ total score and age as independent variables: a higher CTQ total score was associated with PTSD non-remission (OR was .97 with 95% CI [.94-.99], p-value=.011).

Comment#10. 3. Results. The authors mention that 8 were excluded because of ‘incomplete data’. Information on PTSD severity, trauma exposure etc. would be important to describe.

Reply: In our initial database, we stated that 8 patients were excluded because of incomplete data and 50 refused to participate. After a verification in our database, we found that 3 out of these 8 patients initially refused to participate, leading to a number of 53 patients who refused to participate and 5 patients who agreed to participate but who had missing data for at least one questionnaire (versus 50 and 8 before). This is presented in the new study flow chart (page 5). All of these 5 patients had missing data for the PCL-5 (answered less than half of the questionnaires): we were thus not able to compare the PCL-5 scores of these 5 persons with the 298 others. However, we compared these patients regarding other variables, and there was no difference between these 5 persons and the 298 others in terms of age (p=.29), gender (p=.77), AUDIT total score (p=.20), the number of traumatic events experienced (p=.24), and all CTQ subscales (p ranging from .51 to .74). This is now indicated in the manuscript (page 3 line 143 to 145).

Comment#11. 4. Discussion. The authors conclude that “remission varied according to the timing of the trauma (i.e., in childhood)”. This statement is also used in the abstract. I think that this is not entirely accurate and misrepresents what is actually being tested. The authors are not technically examining whether or not traumas that occurred earlier in time are more strongly related to decreased remission, but childhood traumas specifically. Yes, childhood traumas are technically those that occurred earlier in time, but that is not what makes them more ‘traumatic’. The authors should focus on the challenges posed when one experiences a traumatic event (particularly a sexual trauma) in childhood, during a certain developmental stage, not the fact that these events occurred early on in life. In doing so, it suggests that traumas occurred during adolescence or early adulthood would be less severe than in childhood. It’s not about the timing of the event but that these events occurred during a particularly important developmental period in one’s life.

Reply: We also agree with this point, thank you for helping us to improve the clarity of the manuscript. As we did not assess the existence of trauma during adolescence, nor the exact timing of the trauma during adulthood, we cannot determine whether the remission was related to the exact timing of the trauma nor with the duration between the trauma and the present period. We have therefore deleted these sentences in the abstract, discussion and throughout the manuscript, and it has been made clear that remission was related to the existence of childhood trauma rather than to the timing of the trauma. The text now reads:

  • Abstract (page 1 line 33 to 35): “Among patients hospitalized for an AUD with co-occurring PTSD, PTSD remission was more strongly related to the existence of childhood trauma than to AUD or PTSD severity at admission »
  • Discussion (page 13 line 525 to 526): “One of the key findings is that remission varied according to the existence of trauma during a particularly important period in one’s life (i.e., childhood), but not in relation to baseline AUD severity, PTSD severity, or exposure to other types of traumatic event during lifetime”.
  • Discussion (page 13 line 562 to 563): « On the other hand, the lack of association between PTSD outcome and at baseline might suggest that it is not the intensity of PTSD itself that affects outcome, but rather the fact that the trauma was experienced during a particularly important period in one’s life (i.e., a trauma experienced during childhood could lead to earlier changes in personality traits or psychiatric disorders, and be associated with a long-established PTSD that could be harder to treat than a more recent one)”.

We would like to thank you and the reviewers for your work and for taking the time to review and comment our manuscript. We answered your comments to our best and we hope that the modifications proposed improved the manuscript’s clarity and quality.
